**Biological Science Practices**  

ecology, evolution

academic evaluation, h-index, impact factors, academic standards, publishing practices, academic ethics

**Authors for correspondence:**
Colin A. Chapman
e-mail: colin.chapman.research@gmail.com
Nils Chr. Stenseth
e-mail: n.c.stenseth@ibv.uio.no

# Games academics play and their consequences: how authorship, *h*-index and journal impact factors are shaping the future of academia

Colin A. Chapman[1,2,3,4], Júlio César Bicca-Marques[5],
Sébastien Calvignac-Spencer[6], Pengfei Fan[7], Peter J. Fashing[8,9],
Jan Gogarten[6,10], Songtao Guo[4], Claire A. Hemingway[11], Fabian Leendertz[10],
Baoguo Li[4], Ikki Matsuda[12], Rong Hou[2,4], Juan Carlos Serio-Silva[13]
and Nils Chr. Stenseth[9]

[1]Department of Anthropology, Center for the Advanced Study of Human Paleobiology, The George Washington University, Washington, DC 20037, USA
[2]Department of Anthropology, McGill University, Montreal, Quebec, Canada H3A 2A7
[3]School of Life Sciences, University of KwaZulu-Natal, Scottsville, Pietermaritzburg, South Africa
[4]Shaanxi Key Laboratory for Animal Conservation, Northwest University, Xi'an, People's Republic of China
[5]Escola de Ciências, Pontifícia Universidade Católica do Rio Grande do Sul, Porto Alegre, Rio Grande do Sul, Brazil
[6]Viral Evolution, Robert Koch Institute, Seestraße 10, 13353 Berlin, Germany
[7]School of Life Sciences, Sun Yat-Sen University, Guangzhou, 510275 Guangdong, People's Republic of China
[8]Department of Anthropology and Environmental Studies Program, California State University Fullerton, Fullerton, CA 92834, USA
[9]Centre for Ecological and Evolutionary Synthesis (CEES), Department of Biosciences, University of Oslo, P.O. Box 1066, Blindern, 0316 Oslo, Norway
[10]Epidemiology of Highly Pathogenic Microorganisms, Robert Koch Institute, Seestraße 10, 13353 Berlin, Germany
[11]Office of International Science and Engineering at National Science Foundation, Virginia, USA
[12]Chubu University Academy of Emerging Sciences, 1200 Matsumoto-cho, Kasugai-shi, Aichi 487–8501, Japan
[13]Red de Biología y Conservación de Vertebrados, Instituto de Ecología AC, Xalapa, México

CAC, 0000-0002-8827-8140; JG, 0000-0003-1889-4113; SG, 0000-0002-8291-5487;
IM, 0000-0002-0861-7801; NCS, 0000-0002-1591-5399

Research is a highly competitive profession where evaluation plays a central role; journals are ranked and individuals are evaluated based on their publication number, the number of times they are cited and their *h*-index. Yet such evaluations are often done in inappropriate ways that are damaging to individual careers, particularly for young scholars, and to the profession. Furthermore, as with all indices, people can play games to better their scores. This has resulted in the incentive structure of science increasingly mimicking economic principles, but rather than a monetary gain, the incentive is a higher score. To ensure a diversity of cultural perspectives and individual experiences, we gathered a team of academics in the fields of ecology and evolution from around the world and at different career stages. We first examine how authorship, *h*-index of individuals and journal impact factors are being used and abused. Second, we speculate on the consequences of the continued use of these metrics with the hope of sparking discussions that will help our fields move in a positive direction. We would like to see changes in the incentive systems, rewarding quality research and guaranteeing transparency. Senior faculty should establish the ethical standards, mentoring practices and institutional evaluation criteria to create the needed changes.

## 1. Introduction

Being a researcher is a highly competitive profession [1]. This is partially because universities are producing far more PhDs each year than there are academic or

research openings [2]. The profession remains competitive after a researcher obtains a job, because they must pass high standards to obtain tenure and grants. Much of this competition now centres on the ability of researchers to publish in journals with high impact factors. Publishing is an important, admirable and suitable goal; however, this produces a dilemma, as hiring committees, tenure and promotion boards, and granting agencies find it challenging to evaluate the quality of research if it is not in the very specific research area of the reviewer. Evaluators are also extremely busy and look for short cuts in assessing research quality. Furthermore, there has been an exponential growth in both the number of publications and in the number of journals, thereby complicating such assessments [3]. Predatory journals have become a troubling phenomenon, in which almost anything can be published as long as authors pay a fee [4].

To facilitate an evaluation of a researcher in this complex publishing environment, individuals are often evaluated based on their publication number, the number of times they are cited, the impact factors of the journals they publish in and their $h$-index. A journal's impact factor is defined as the year's average number of citations per paper published in a specific journal during the preceding 2 years [5]. An individual's $h$-index considers the researcher's best-cited papers and the number of citations that they have received [6] (i.e. an $h$-index of 10 indicates an author has 10 papers that have each been cited 10 times or more).

Scores for these indices have great consequences. For example, in some countries and disciplines, publication in journals with impact factors lower than 5.0 is officially considered to have no value [7,8]. Furthermore, it is not uncommon to hear that the only articles that count are the ones in journals with an impact factor that is above an arbitrary value, or, worse, that publishing in lower-tier journals weakens CVs [9]. It is surprising that the $h$-index now plays such an important role as it was only in 2005 that Jorge Hirsch proposed it as a tool for quantifying the scientific impact of individuals [6]. The situation is made more significant because the same metrics are often used inappropriately by universities, journals, granting agencies and reviewers. For example, a reviewer may look up the $h$-index of the person for whom she/he is reviewing a grant when the granting agency requests instead they evaluate only the proposal.

Even though there is an extensive body of literature criticizing these indices and their use [7,10], they are still being widely used in important and career-determining ways. As a result, it is important to consider their meaning and the consequences of their use, particularly for young scholars. There have been many appeals from the academic community to not judge applications for positions, tenure or grants by the number of publications or quantitative indices, like h-indices, but rather to focus on the importance of achievements [3,11,12]. We agree that this would be ideal, but given the increasing demands placed on university professors [13] and the decreasing support professors have from their universities [14], it is unrealistic to assume this ideal will be met. Furthermore, as with all indices, people can 'play games' to better their scores. Such gaming has resulted in the incentive structure of science increasingly mimicking economic principles, but instead of the incentive being monetary gain, it is with respect to increasing one's score [5]. This raises serious ethical questions as to the appropriate goals of an individual's research.

Ultimately, what hiring committees, tenure review boards and granting agencies are trying to do is measure the quality of research; unfortunately, this is very hard to do. Quality research must involve solid rationale, theoretical framework, and methods, appropriate statistics (if applicable), sound logic and proper citation of the literature. But meeting these standards is insufficient to be viewed as high quality. A study with all of these characteristics, but that repeats what is well established by many previous studies, offers only confirmatory, not ground-breaking findings. Reproducibility, replicability and reliability all have a role in quality research. However, some research in, for instance, ecology and evolution is difficult or impossible to replicate with reliability. Repeating studies previously conducted at one location and point in time are difficult to interpret because differences found could be due to spatial or temporal variation in some unmeasured variable. Quality research must be novel, creative and positively influence the development of a field. In many fields, this can only be assessed by a very restricted pool of people and even these scientists may not recognize truly innovative ideas that go against current thought. Such research may only be recognized as being influential a decade or more after it is published. For example, Hamilton's 1964 paper [15] dealing with inclusive fitness and the evolution of social behaviour is widely acknowledged as one of the most significant extensions of natural selection [16]; yet this paper's importance was not recognized until it was popularized by E. O. Wilson's book on sociobiology that was published in 1975 [17].

Calls to mobilize reform in the tools, practices and study of the assessment of science and scientists gained momentum this decade, particularly through the San Francisco Declaration on Research Assessment (DORA) [18]. Published in 2013 and now signed by some 15 000 individuals and 1550 organizations worldwide, DORA recommends that journal impact factor not be used as a surrogate measure of an individual research article, to assess an individual scientist's contribution, nor in hiring, promotion or funding decisions. Steps that individuals, universities, funders and scientific societies can take to stop the mismeasure of the quality of research output are illustrated by a growing list of good practices (see https://sfdora.org), such as the French National Research Agency's awareness and training programme announced with the 2019 call for proposals. Use of improved indicators by all stakeholders in the system is echoed in the 10 principles of the 2015 Leiden Manifesto, which includes the role of qualitative judgement and contextualization of metrics for the research field, and other international or national efforts [19]. The Metric Tide report (2015) makes 20 recommendations specific to improve research assessment in the UK system; however, the dangers raised that journal impact factor and citation counts can be gamed are universal [20]. More recently, after reviewing 22 documents impacting clinicians and life scientists, Moher and colleagues [21] outlined six general principles about what to assess, how to assess it, the need for complete and transparent publication, the need for openness of data and results, the need for research on new and existing assessment criteria, and the importance of rewarding intellectual risk-taking to encourage ground-breaking research. Despite the groundswell of calls for the transformation of the assessments and incentive structures, systemic change is difficult and slow. Implementation of any of these recommendations and principles will be out of pace with researchers' current career paths.

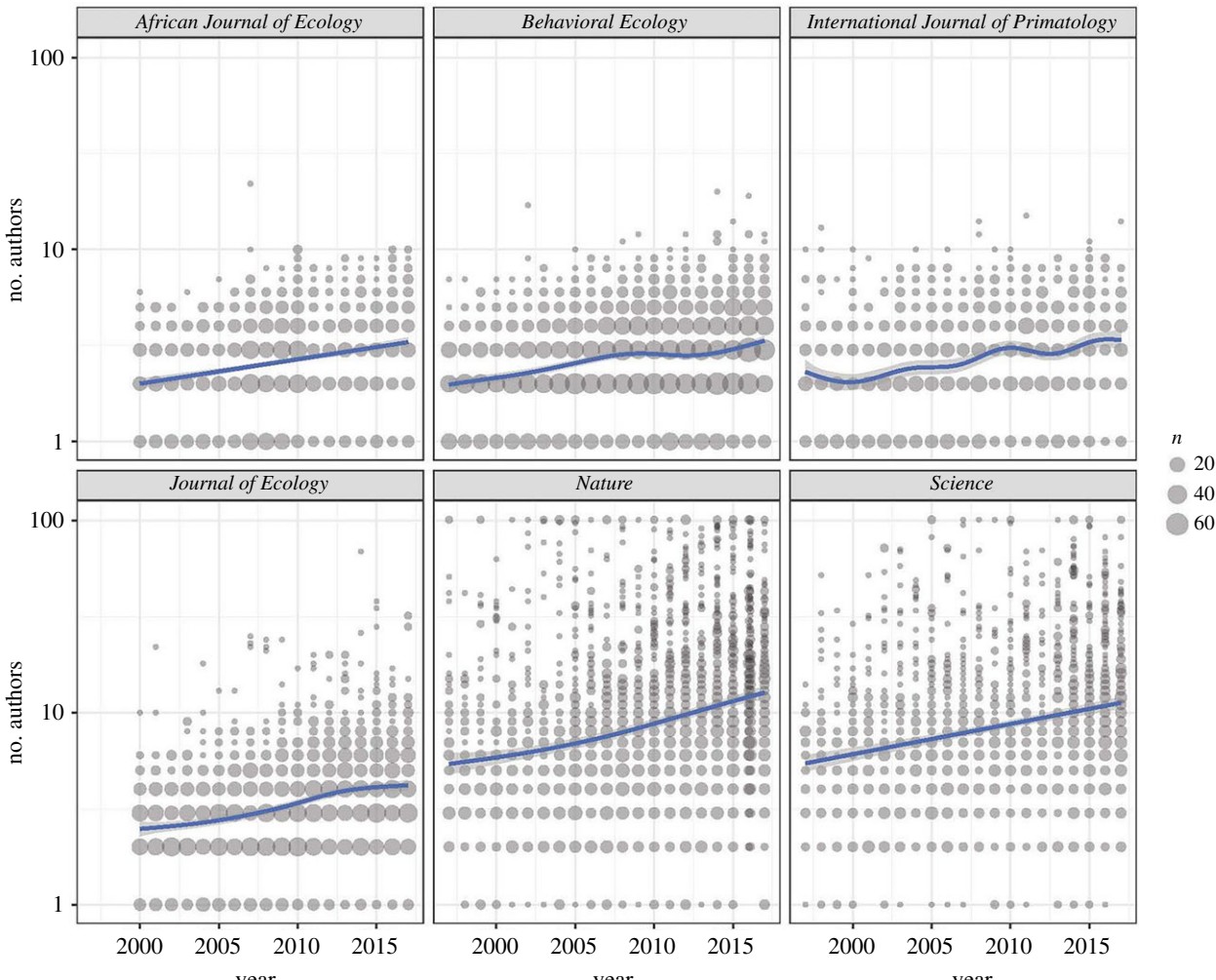

**Figure 1.** The number of authors of research articles in six journals through time. The area of each circle corresponds to the number of publications with that publication number for that year. To aid in the visual interpretation of the data, a generalized additive model was fitted to the data. For ease of interpretation, the number of authors is truncated at 100, meaning that publications with more than 100 co-authors are plotted here as just including 101 co-authors.

Given this current situation, the first objective of our paper is to evaluate how authorship, *h*-index of individuals and journal impact factors are being used and abused. Second, we speculate on the likely consequences of their continued use with the hope of sparking discussions that will shape our fields to move in a more positive direction. Considering academic careers as a game (in which agents' decisions are informed by the above metrics) implies that all agents can try to affect the game's outcome. We are all aware that an individual's ethics and social milieus play an important role in the academic game, which we address in passing in the different sections and more specifically in the conclusion of this article. To ensure a diversity of perspectives and experiences in our evaluation, we gathered a team of academics from around the world (Brazil, Canada, China, France, Germany, Japan, Mexico, Norway, Uganda and USA), in different disciplines and at different career stages.

## 2. How authorship and indices are being used and abused

In the last two decades, authorship practices have changed dramatically. In many fields, papers were traditionally single authored or involved only a few authors. Increasingly, multi-authored articles are now the norm in most disciplines.

In the biomedical field, the number of authors on publications has increased by approximately one author per article per decade and high-impact journals have longer author lists than journals with lower impact [22]. The trend for an increased number of authors on papers is evident in all types of journals, be the journal regional, taxonomic, theoretical to a subfield or the most general scientific journal (figure 1). This partially represents science becoming more multi-disciplinary and that researchers with different skills and tools are needed to address the question of interest. However, the multi-author trend also facilitates developments that are not so positive; specifically, including authors to game the system to increase the number of papers an author publishes.

It is widely acknowledged that there are serial abusers of authorship etiquette involving academics in positions of authority who carefully distort authorship credit to their benefit [23]. A junior author has little option than to submit to the authority figure. However, authorship manipulations are not always negative for the individuals involved. In fact, it has become increasingly common for senior researchers to put their students' and post-docs' names on publications where they do not deserve authorships, to help them obtain academic positions. This practice benefits the student, as it presumably helps them acquire positions, and benefits the professor, as they can claim that they are successful mentors, but raises ethical concerns if not all researchers participate in this

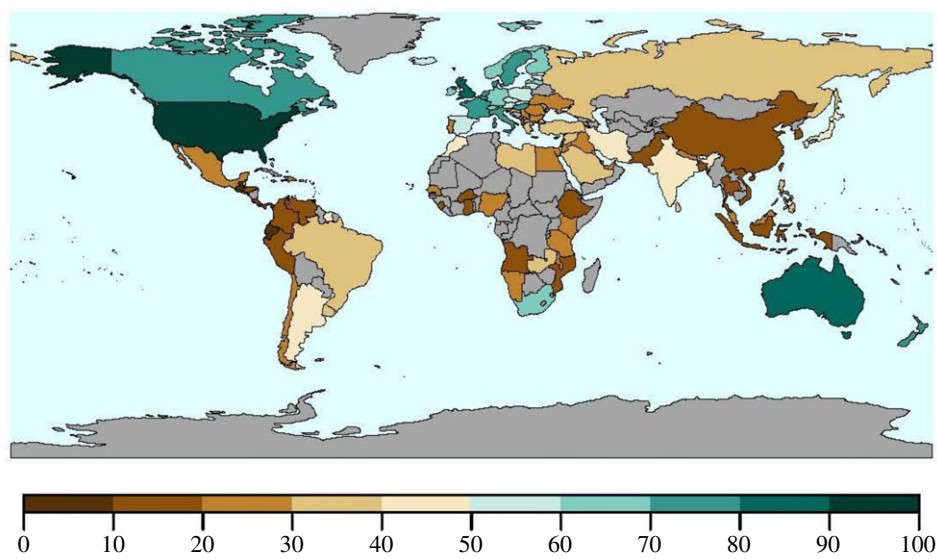

**Figure 2.** The individualistic score based on Hofstede's cultural dimensions theory (this is the amount to which people in a society/culture are integrated into groups) of countries for which scores are available (https://www.hofstede-insights.com/product/compare-countries/). The higher this score, the greater the preference for a loosely knit social framework in which individuals are expected to take care of only themselves and their immediate families; the lower the score, the greater the preference for a tightly knit framework in a society where people expect their relatives or members of a particular group to look after them in exchange for unquestioning loyalty. Countries with a lower score are thought to be more likely to include unmerited authors on publications [26]. Countries in grey do not have available data. (Online version in colour.)

manipulation in similar ways. Success in mentoring is an evaluation criterion used by some granting agencies (e.g. Natural Sciences and Engineering Research Council of Canada—NSERC). Professors also play a strategy involving 'you scratch my back and I will scratch yours', with the simple expectation that if they put a colleague on a publication when they contributed little, the favour will be returned [24]. Furthermore, there are strategic reasons for manipulating the author list. For example, in the medical field, papers with at least one author from a basic science department are more likely to be published in a high-impact journal than papers from only clinicians [25]. There is also a cultural basis to unmerited authorship which appears to be influenced by how individualistic or collectivistic a society is [26]. Researchers from countries that have a more collectivistic perspective (the practice of giving a group priority over each individual) are more likely to add unmerited co-authors than scientists from individualistic countries [26]. Based on Hofstede's cultural dimensions theory [27], the individualistic score has been determined for a large number of countries, allowing a general evaluation of trends (figure 2).

From the experiences of some of the co-authors of this paper, when researchers are conducting field research in foreign countries, it can be strongly encouraged that a member of the permit agency be included as an author on the resulting publications. Some granting agencies (e.g. National Geographic) require or strongly encourage that researchers from the foreign country be included in the grant and publications. This appears to be an admirable strategy aimed at improving host-country science. However, host-country inclusion on publications, when there is no appropriate involvement or training, overemphasizes the skills in the host country and stands in the way of calls for enhanced training efforts [28]. There is a variety of other sorts of inappropriate manipulation of author lists. In the most extreme cases, researchers have paid to be an author on papers that they did not contribute to. Hvistendahl [29] reports a company selling co-first-authorship for US$14 800.

There are efforts to control the manipulation of author lists, such as journals' requirement of explicitly disclosing each author's contribution to the manuscript. However, given the intense competition for publications, it is very likely that those inappropriately including people on author lists will simply 'adjust' their statements to fulfil the expected contribution.

## 3. *h*-index, citation numbers, number of publications, etc.

An individual's previous publication and grant record are often evaluated relative to others; thus funding and other opportunities rely heavily on how an individual ranks relative to the competition [5]. We examined the review criteria used by government granting agencies for the countries that the authors of this article are from to assess the importance of these metrics. In general, if referenced, these statements do not set specific cut-off points or similar criteria, and thus the decision to rely on metrics, like the *h*-index, is left to the reviewer's discretion. In fact, NSERC (Canada) states that the impact of the researcher does not refer to quantitative indicators such as the impact factor of journal or *h*-index. By contrast, the Brazilian National Research Council (CNPq, as well as CAPES) use one or a combination of metrics, including journal impact factors, number of publications in indexed journals and *h*-index to rank researchers for granting fellowships. In South Africa, the National Research Foundation asks for applicants' *h*-indexes and the University of KwaZulu-Natal is implementing a ranking system for journals that determines the allocation of university incentive funds.

Also relevant to the use of the summary metrics are the restrictions of granting agencies on what can be submitted as a biographic record. Most granting agencies have length limitations on CVs (e.g. Deutsche Forschungsgemeinschaft in Germany, the Swiss National Science Foundation and the

National Science Foundation of China request five recent publications; the National Science Foundation of the US has a 2-page maximum; and the Canada Common CV, used for many grant programmes, only permits reporting of publications in the last 6 years). These rules partially reflect the desire not to burden reviewers with extremely long CVs. However, this encourages the use of summary metrics, such as the $h$-index, as there is no other way to convey an extensive research career.

Such practices exist, although confusion is rapidly growing around what constitutes a high rank with respect to the different indices. For example, it is a common belief that the only articles that count are ones in high-impact journals [9]. However, not all papers published in journals with a high impact factor are heavily cited [30]. For example, 89% of *Nature*'s 2004 impact factor of 32.2 was accounted for by only 25% of the papers [31]. Furthermore, researchers often believe that given that time and effort are limited, they can either produce a few high-quality papers or subdivide datasets in an effort to produce many papers of lower quality. The opinion has recently swung dramatically in favour of a few high-quality papers. Despite this perception, research indicates the opposite relationship, in which publishing more papers is associated with increased citation accumulation [32]. This suggests that an individual's citation counts, and the rewards that are associated with them, can be more stochastic than previously believed (i.e. if you publish more papers, there is a greater chance that one will be highly cited by chance [32]), and of course there is increased opportunity for self-citation.

Self-citations are another area of confusion around the indices used to evaluate individuals. Some sources of citation history provide information on self-citation, while others do not (e.g. typical Google Scholar searches do not, nor does the Researcher ID in Publons). But how should self-citations be considered? Research should be a process of intellectual growth where one study leads to the next. From this perspective, citing oneself should be considered as a natural and acceptable procedure. However, scientists also cite themselves to increase their citation scores. A study of 45 000 citations produced in Norway found that 36% of all citations in the first 3 years after publication represent self-citations [33]. This study found that the highest share of self-citation was among the least cited papers, and that there was a strong positive correlation between the number of authors of the publications and the number of self-citations.

When evaluating individuals, many other issues need further consideration. For example, if indices are to be used, what index should be used and from what source? An individual's $h$-index can differ dramatically depending on its source; for example, Google profiles typically report higher $h$-index values than the Publons (the former Web of Science) database, probably because Google profiles include books and book chapters, while Publons does not. Furthermore, there are a large number of indices proposed and it is clear that some indices will produce biases that favour some groups over others [34]. The $h$-index depends on the researcher's age. To account for this, a person's $h$-index can be divided by their scientific age (the number of years since an author's first publication) to generate an $m$-index, which is viewed as the speed with which a researcher's $h$-index increases [6,35]. However, the $m$-index for young scientists can be dramatically affected by a year off for parental leave and a 1-year difference in the onset of publishing, especially impacting women scientists

(e.g. 1.33 versus 2.00 for 8/6 years versus 8/4). The use of the $m$-index also disadvantages students publishing early during their undergraduate studies.

Publication databases also produce different results; which should be used? Some indices include variables like how often a scientist asks and answers questions in online forums or their number of followers (e.g. ResearchGate). The Altmetric score for a publication evaluates how often a publication has been referred to in news stories, blogs, Twitter, Facebook and Reddit, providing insights into public interest. Should such metrics be considered when judging the merits of a publication or a scientist? Altmetric scores are a way to evaluate the broader impacts of research. National Geographic provided these scores to the Committee for Research and Exploration. In 2016, *Primates* started a new annual prize for the paper with the highest Altmetric score and full-text downloads. The National Science Foundation of China announces information on the top 100 papers reported by Altmetric. So it appears that the question is not whether scores such as these should be used—they are already being used. The question becomes how they should be used and how we can ensure that they accurately reflect what academia wants them to reflect. Altmetric scores can easily be increased simply by the journal, university or author sending out tweets or Facebook postings about newly published articles. Some journals have begun to make a Twitter text line obligatory for final submissions. Is such self-promotion a strategy that the academic community is selecting for?

## 4. The uses and abuses of journal impact factors

Journal impact factors are important because of the ways the academic community has come to rely on them. It is widely acknowledged that journal impact factors should not be used to rank individuals. This stems from the fact that not all papers published in journals with a high impact factor are heavily cited [30,31]. As discussed above, this was formalized in 2012 with the writing of DORA [7,36].

Despite such declarations, journal impact factors continue to be widely used to assess the value of individual scientists with negative effects on individuals and on academic society. For example, these practices compel researchers to first submit their papers to very high-impact-factor journals and following rejection progressively circulate the manuscript through journals with lower and lower impact factors [11]. This practice of progressive submission down the ladder of journal impact factors can be particularly detrimental to academics just entering the job market. Their advisers can coerce them to submit to the highest-impact journals, which has little cost to the adviser. This is very beneficial for the student if successful, but to the young scholar, progressively circulating down the tiers of journals can be discouraging and stressful, and if they are not successful at getting published in an upper-tier journal by the time they apply for positions, it may mean they are unsuccessful in the job competition. This raises important questions concerning the appropriate ethics of mentorship.

Academics should keep in mind that journals are, for the most part, a for-profit business, with profit margins reaching as high as 40% [37]. In fact, in 2017, the global revenues from scientific publishing were estimated to be £19 billion (approx. US$24 billion), and in 2010, its profit margins were

higher than Apple, Google or Amazon [37]. The need for profit can surpass doing what is best in an academic or societal sense. A major way journals manipulate their impact factor is by publishing in areas with the largest numbers of researchers (e.g. cancer research) and in areas where research outputs are the greatest. Article output varies significantly among disciplines; researchers in the 'hard' sciences (physical and biomedical sciences and engineering) publish approximately 2.5 articles per year, the social sciences publish 1.7 articles and the arts/humanities is the discipline publishing the least (under 1) [38]. There are other ways journals can manipulate their impact factor. One estimate found that 20% of the researchers who responded to a survey had experienced incidents where an editor had asked them to add citations to a submitted article from the editor's journal even though the editor did not specify a specific article or topic [39]. The same study found that 57% of the researchers added superfluous citations to their papers, citing the journal that the manuscript was submitted to in an attempt to increase its chance of publication. More alarming is the fact that editors have organized citation cartels where a group of editors have recommended authors to cite articles from each other's journals [7,40].

Journals can increase their impact factors in subtler ways. The reason that many journals no longer publish notes is largely because they have lower citation rates than articles; yet, notes provide valuable information for reviews and meta-analyses. Editors can also manipulate their journal's impact factor by publishing review articles and meta-analyses that are cited more often [41], rejecting negative or confirmatory studies, attracting articles written by large research groups or with a large number of authors, or those written by renowned scientists and leaders of research, regardless of the real quality, or by publishing articles that deal with 'hot' topics [42].

## 5. Possible consequences of relying on the *h*-index and journal impact factors

In the last couple of decades, the academic landscape has changed dramatically, and evidence points to an accelerating rate of change. We echo the statements of many scientists and call for the elimination or at least the appropriate use of metrics to evaluate individual researchers, research institutions and journals [7,23,24,30]. A promising recent development in Canada is that all major funding agencies have now signed DORA and a national discussion of research excellence is under way. However, despite major academic societies making such calls [7,36], it is our opinion that the inappropriate use of metrics is a major influence on scientists today.

Currently, there is strong pressure on individuals, particularly young researchers, to play the game to advance their scores. If one does not, and there are no available alternatives, one runs the strong possibility of bringing up the rear and being excluded from an active research career. This reality has led to pleas asking for 'more clarity on what the game actually is' [9]. At the very least, enough clarity must be provided so that researchers can make their own professional decisions on how to use their own strengths to advance their careers and so they can decide if they will join the call for the elimination of metrics.

With increasing demands being placed on universities, granting agencies and researchers, there will be an increased pressure to use short cuts, such as metrics, rather than

investing in expert review [43]. In fact, some granting agencies (e.g. National Geographic) have scaled back on their reliance on expert review as costs mount and submissions increase. A recent analysis by Eyre-Walker & Stoletzki [44] suggests that post-publication peer review is prone to error, biased by the reviewer's perception of journal impact factor and expensive. This leads them to question the use of expensive expert review in post-review assessments and to suggest journal impact factor is more appropriate. They find a lack of agreement among reviewers and interpret this to indicate that these assessments are not reliable. However, the authors fail to recognize that this lack of agreement may reflect that reviewers often assess a different aspect of research or may be considering a work's merit to different sub-disciples [45]. The pressure to use metrics as short cuts and the recognition that each metric has its limitations have called for assessments using multiple metrics (e.g. number of views, researcher bookmarking, social media) [45], and these metrics, particularly social media, have been promoted by publishers and universities. Altmetric is a one such metric that is gaining prominence. It provides a score of the online attention received by research outputs based on social media (e.g. Twitter and Facebook), traditional media, blogs (both from institutions and individuals) and online reference managers. While improving metrics may be helpful to evaluate some aspects of research life, we strongly recommend that they should not be rewarded with perverse incentives, nor should the metric be easily gamed. Altmetric scores can be gamed by posting more on social media and having your friends post, or even writing programs to post or repeatedly download your articles. Furthermore, rewarding individuals with high scores on such metrics selects for particular types of scientists. The use of social media in academia must be made with caution so that measuring the traces of research impact does not become the goal, rather than the quality of the research itself [8]. We encourage, rather than developing other metrics, that academia improves systems of expert review to ensure quality. This could include having reviews shared and discussed among evaluators and this discussion being considered in the publication and grant process, and having reviewing receiving more importance in the tenure and promotion process. Established researchers should encourage the appropriate use of existing metrics and insist that they are never the sole—or even the major decisive factor—in an evaluation. They should also evaluate and potentially promote alternative avenues of transparent publishing, such as Faculty of 1000, where articles are first shared as preprints and then peer reviewed by invited referees whose names and comments are made available on the site, or arXiv and bioRxiv, which are repositories of electronic preprints that are approved for posting, but not peer reviewed, and then often submitted to journals. Action is being taken on some issues, but we see a critical need to encourage a more proactive discussion. As a result, we have considered the pressures on researchers and journals that we have outlined above and *speculated* on how they will change funding, publications, academics in general, and the traits of a successful researcher (box 1). By making such speculations, we hope to spark a discussion that will shape the future of academia and move it in a more positive direction. Scientists represent some of the most creative minds that can address societal needs. Is it now time we forge the future we want?

**Box 1.** Speculations regarding the consequences of allowing current evaluation metrics of individuals and journals to continue. By reporting these speculations, we hope to stimulate discussion regarding how scholars should plan to shape the future of academia.

**Publications**

— As review articles and meta-analyses are more frequently cited than data papers, they may increase in frequency and there will be an increase in the 'mining' of existing, often long-term, datasets, at the expense of getting new field data.

— Books and edited volumes, which are harder to evaluate and generally thought to be less critically reviewed, may become less valued in the future, differentially affecting fields.

— Lower-impact and very specialized, often society, journals from developing countries may receive less and lower-quality submissions putting them into a vicious circle that could lead to their termination. This may be particularly serious for non-English journals. It is important to raise the topic of language because it is the single most important issue that pulls non-English speaker scientists down in these ranks.

— The peer-review system will increasingly become burdened as researchers submit to one high-impact journal after another and journals will have a progressively harder time finding reviewers as a result of 'reviewer fatigue'. This may result in the increase in time from the first submission to publication and a decrease in publication quality as reviewers are over-worked.

— The body of scientific literature will become increasingly diluted with poor-quality publications, because the main incentive behind publications is the status authors can derive from them, rather than knowledge gain. Using the body of scientific literature will require increasing skills to read large amounts of text, and prudent use of logic, plausibility and parsimony on the part of the reader to separate valuable publications from invalid ones, as indicators like journal status become less reliable.

— With the growing number of co-authors, the incentive to neglect good scholarship increases, as the putative blame is shared among many. Again, critical reading becomes essential for the evaluation of publications.

**Academics**

— Sex biases will persist and be promoted by some metrics. Female scientists produce fewer papers than males [46], which affect their $h$-index; parental leave negatively affects their $m$-index; and assessors may be unaware of the biases they contain [35].

— Increased stratification of academia—the difference between the 'haves' versus 'have nots' will increase [24].

— University and granting agencies that reward the training of many graduate students, rather than encouraging high-quality mentorship, will maintain the hypercompetitive system that encourages gaming and allow universities to continue the abusive system of hiring sessional teachers. In 2013, part-time or adjunct professor jobs made up 76% of the academic labour force in the US and were paid $2700 per class [1].

— There may be a decreased effort to work with students from low-income countries, particularly non-English-speaking countries. Such students often have poorer training when entering graduate programmes and thus it often takes them longer to complete degrees, they cost more and they are more likely to publish in regional, low-impact journals [28].

— As such games become progressively known by the general public, they will have less respect for universities and the findings of scientists.

**Financial implications**

— Funding may decrease for fields that traditionally have low impact journals and citation histories. While recognizing differences among fields, this can impact whole disciplines, such as the arts, and subfields within traditional departments (e.g. classical ecology versus genomics and other -omics fields within biology departments).

— Since universities are increasingly looking to receive funding from grant overhead, this might provide incentives to reduce their funding to departments that do not generate significant overhead, reducing the diversity of science and the student offerings.

— If institutions are financially rewarding individuals based on such metrics, it will increase the importance of such metrics, even if national and international agencies move away from their use.

**These games will select for the following traits in successful individuals**

— People who aggressively promote themselves will do better in an academic competition, so more time will be spent on promotion, using avenues like Facebook, Twitter, webpages, email lists and online blogs. Since time is limited, this will result in less time for research and training.

— There will be a selection for academics in positions of authority who carefully distort authorship credit to their benefit, often at the expense of graduate students and post-docs.

— Graduate students and post-docs might be encouraged by their mentors to publish more review papers and meta-analyses, as they receive more citations. When mentors make such recommendations, the costs to themselves may be small whereas the benefit may be large, but to the junior author, if these papers are rejected, the cost may be huge.

— Joining prestigious multi-authored teams will be encouraged.

## 6. Conclusion

In the end, ethics lies at the very heart of scientific endeavours and much of our work revolves around ethical considerations.

Researchers are used to thinking about and discussing ethical matters; we can build on that and extend this tradition to reflective action on the way we construct our communities. This leads us to structural considerations. First, since all researchers have

the possibility to reach the highest positions in the scientific community, there is the real possibility of educating our own students in the proper use of performance metrics, which will ultimately change the academic landscape. Second, scientific communities are often relatively free from political influence (i.e. we can write on the topics we want to as long as it represents good science), so we think scientists should abandon their position of neutrality and commit to advocating for the implications of their science; however, this will only be possible if we maintain a high ethical standard on how science is evaluated.

Changing the academic culture to have a more just means of evaluation of individuals and journals will come about only if there are changes in the incentive system that drives so much of academia. The system is currently heavily weighted towards rewarding faculty for research output in the form of high-ranking publications. Change should start in institutions of higher education where more rewards should be given to students for the highest quality of research and faculty for the highest quality of mentoring [47], where quality is judged on the basis of narrative expert evaluations, rather than indices. This should be accompanied in the tenure and review process where senior faculty should take the time to evaluate the quality and guarantee transparency. Outside of the home institution (e.g. publishing and grant reviews), reviewers are needed who can do the critical synthetic thinking, truly evaluate quality across and within disciplinary boundaries, and ensure that quality remains the gold standard.

Funders must also play a leading role in changing academic culture with respect to how the game is played. First and foremost, funders have a clear role in setting professional and ethical standards. For example, they can outline the appropriate standards in the treatment of colleagues and students with respect to such difficult questions as what warrants authorship and how to determine its ordering. Granting agencies should clearly emphasize the importance of quality and send a clear message that indices should not be used, as expressed by DORA, which many agencies have endorsed. Of particular importance is for funders not to monetize research outputs based on metrics, such as the $h$-index or journal impact factor. Monetization largely based on such metrics is being done in many countries, such as Australia, China, Mexico, Scandinavia, South Africa and Uganda, and incentives can be as high as US$165 000 per publication [19,48]. At all levels, a large proportion of the responsibility must fall on senior faculty to be role models expressing the highest ethical standards, being superior mentors of only a few students, making the effort to make reviews based on quality and establishing institutional evaluation criteria. One must keep in mind that senior faculty probably hold their current positions through their success in the game, which may or may not have been achieved by using the most ethical ways. Thus, institutions must also play a role by training mentors to create a healthy and ethically robust culture and encouraging team mentorship where deviations from the highest professional standards can be monitored and, where necessary, appropriately disciplined. The system will not change unless faculty behaviour changes, and changing the incentive system is critical in that regard.

Data accessibility. Data for figure 1 available from the Dryad Digital Repository: https://doi.org/10.5061/dryad.fn2z34tpx [49].

Authors' contributions. C.A.C., C.A.H. and N.C.S. came up with the original idea of writing this paper. C.A.C. wrote the first draft and all authors contributed examples from their home country or from the country with which they work (C.A.C., Canada, USA, Uganda, global national comparisons; J.C.B.-M., Brazil; S.C.-S., France, Germany; P.F., China; P.J.F., USA, Ethiopia; J.G., Germany, USA, Canada; S.G., China; C.A.H., USA, Madagascar, global national comparisons; F.L., Germany, Côte d'Ivoire; B.L., China; I.M., Japan, Borneo; R.H., China, Uganda; J.C.S.-S., Mexico; N.C.S., Norway, Ethiopia, global national comparisons). R.H. collected the data for figure 1, and J.G. did the analysis and created this figure. C.H. states that all opinions, findings and conclusions expressed in this material are those of the authors and do not necessarily reflect the views of the National Science Foundation.

Competing interests. No authors have a conflict of interest.

Funding. C.A.C. was supported by the Humboldt Foundation, NSERC and the Robert Koch Institute while writing this paper and he would like to highlight that the IDRC grant 'Climate change and increasing human-wildlife conflict: how to conserve wildlife in the face of increasing conflicts with landowners' facilitated discussions that gave rise to this manuscript. This study received support from the National Natural Science Foundation of China (grant nos. 31730104, 31870396).

Acknowledgements. We thank Marcus Clauss, Finn-Eirik Johansen, Marcia McNutt, John-Arne Røttingen, Dipto Sarkar and Tore Wallem for comments on an earlier version of this contribution.

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
