## [Reviewer comments · Proceedings of the Royal Society B: Biological Sciences]

Review History

RSPB-2019-2047.R0 (Original submission)

Review form: Reviewer 1

Recommendation

Accept with minor revision (please list in comments)

Scientific importance: Is the manuscript an original and important contribution to its field?

Good

General interest: Is the paper of sufficient general interest?

Good

Quality of the paper: Is the overall quality of the paper suitable?

Good

Is the length of the paper justified?

Yes

Should the paper be seen by a specialist statistical reviewer?

No

Do you have any concerns about statistical analyses in this paper? If so, please specify them explicitly in your report.

No

It is a condition of publication that authors make their supporting data, code and materials available - either as supplementary material or hosted in an external repository. Please rate, if applicable, the supporting data on the following criteria.

Is it accessible?

No

Is it clear?

Yes

Is it adequate?

Yes

Do you have any ethical concerns with this paper?

No

Comments to the Author

The topic of this MS is well-suited to the new 'Biological Science Practices' section of the journal. It deals with the ways in which scientists and institutions have reacted, and continue to react to the use of bibliometric indices in assessment of quality. The piece is very well written. It paints quite a bleak picture of widespread 'gaming' of the system and argues that this is detrimental to the quality of scientific output and to the careers of many promising young scientists.

The arguments here are familiar and most of the article is devoted to reviewing multiple previous studies on the operation and failures of indices like the Journal Impact Factor and the h-index.

The only new data come from a survey of trends in numbers of authors per article in leading specialist and generalist journals. The upward trends are worrying but the authors are unable to separate legitimate causes (such as increasing collaborative working across teams) from gaming by inappropriate addition of authors.

There is no reference to Eyre-Walker and Stoletzki (2013). I think this would be a useful paper to consider because of its comparison between expert assessment of the quality of papers and assessment based on indices. It actually questions the assumption that expert assessment is preferable, by showing that the correlation between assessments by different experts is low. This is relevant to the recommendation to that more time and effort should go into critical reading of output, rather than relying on indices, for example to compare applicants for positions.

The authors conclude by placing the burden on senior faculty to change their own behaviour and instil better attitudes in the young scientists they train. Such a change would certainly be a good thing but one must keep in mind that senior faculty hold their current positions through their success in playing the game: they may be the least likely group to see a strong motivation for changing behaviour. On the other hand, there is strong evidence that institutions (particularly funding bodies) do have the power to change behaviour. This is very clear from the way UK academics and universities have adjusted their priorities in response to the RAE/REF. This has not always pushed in the right direction but the push has certainly been effective.

I see this MS as a worthwhile contribution to an ongoing debate. Most readers will already be aware of the main points raised: this is not a new discussion. The new data confirm a trend that is familiar anecdotally and they form quite a small part of the MS. Perhaps the conclusion, with its call for action, is the most valuable element.

Eyre-Walker, Adam and Stoletzki, Nina (2013) The assessment of science: the relative merits of post- publication review, the impact factor, and the number of citations. PLoS Biology, 11 (10). e1001675. ISSN 1544-9173

Review form: Reviewer 2

Recommendation

Accept with minor revision (please list in comments)

Scientific importance: Is the manuscript an original and important contribution to its field?

Good

General interest: Is the paper of sufficient general interest?

Good

Quality of the paper: Is the overall quality of the paper suitable?

Good

Is the length of the paper justified?

Yes

Should the paper be seen by a specialist statistical reviewer?

No

Do you have any concerns about statistical analyses in this paper? If so, please specify them explicitly in your report.

No

It is a condition of publication that authors make their supporting data, code and materials available - either as supplementary material or hosted in an external repository. Please rate, if applicable, the supporting data on the following criteria.

Is it accessible?

N/A

Is it clear?

N/A

Is it adequate?

N/A

Do you have any ethical concerns with this paper?

No

Comments to the Author

This is an interesting and informative discussion of the well-known problem in modern research of gaming publication metrics. It is not particularly original – this is well-trodden ground – but there seems to be value in laying these arguments before the Ecology and Evolution community, who appear to be the primary target.

The article provides an interesting and well-structured discussion of the main issues. In a few places there seemed to me to be some gaps in consideration of the literature but that perhaps reflects my UK-centric perspective. I think this is a useful contribution to an important and growing discussion.

I have some minor comments that I think the authors should consider. I have followed the authors' page numbering.

Page 2: "much of this competition now centers on the ability of researchers to publish in high quality journals." I would question the use of the term 'high quality' in this context. Certainly, there is pressure to publish in journals with high impact factors and/or established brands, but these terms have a complicated relationship with 'quality'. Arguably there are some very high-quality disciplinary journals that do not have very high impact factors. Perhaps this could be explored?

Page 3: the statement "an extensive body of literature" is supported by a single reference to a one-page editorial in Science. I think the authors could do more to cite the primary literature selected from this body of evidence.

Page 4: One of the strengths of the paper is the wide range of nationalities of the authors. I think it would be helpful to readers to list these in the body of the text so it is clear what geographical contexts/experiences are being discussed.

Page 5 (and elsewhere): reference is made to the "h factor" but there is no such thing. The metric in question is the "h index".

Page 7: Some of the difficulties associated with the use of alternative metrics were discussed in some depth in The Metric Tide report (2015). See especially the supplementary literature review: <https://re.ukri.org/news-events-publications/publications/metric-tide/>

Page 8: Rather oblique reference is made to the Declaration on Research Assessment (DORA). This is incorrectly described as an initiative of the American Society for Cell Biology. Although the ASCB were certainly involved, so was EMBO and so were other stakeholders. More up-to-date information on the number of organisational signatories (now over 1500 – not just 75!) can be found at <https://sfdora.org> (e.g. see <https://sfdora.org/signers/>)

Page 8: Publisher profit margins are reported as being "as high as 26%" but I have seen even higher reports (more like 35%) e.g.

<https://www.theguardian.com/science/2017/jun/27/profitable-business-scientific-publishing-bad-for-science>; <https://journals.uic.edu/ojs/index.php/fm/article/view/4370/3685doi:10.5210>

Page 9: "we see a critical need to encourage a more proactive discussion". I agree but think it may be useful to discuss recent initiatives by the Wellcome Trust and the linkage in Plan S of the drive for open access with necessary reform of research evaluation.

Page 9: The question of researchers' ethics is raised here for the first time (despite 'ethics' being a keyword for the article). But the issue – and I agree it is an important one – is barely discussed. There is a fundamental disconnect, I think, between the ethical impulses that bring many people into a career in research and the mechanisms by which they are evaluated. This seems to me to be one of the root causes of the increasing concerns about reproducibility. I think this topic could be explored in more detail – but perhaps it is not such a big issue in ecology and evolution?

Legend to Fig. 2: It would be helpful to explain what the 0-1 (or is it 0-100?) scale means. I am not familiar with Hofstede's cultural dimensions theory.

Decision letter (RSPB-2019-2047.R0)

14-Oct-2019

Dear Dr Chapman:

Your manuscript has now been peer reviewed and the reviews have been assessed by an Associate Editor. The reviewers' comments (not including confidential comments to the Editor) and the comments from the Associate Editor are included at the end of this email for your reference. As you will see, the reviewers and the Editors have raised some concerns with your manuscript and we would like to invite you to revise your manuscript to address them.

Research ethics:

Use of animals and field studies:

It is a condition of publication that you make available the data and research materials supporting the results in the article. Datasets should be deposited in an appropriate publicly available repository and details of the associated accession number, link or DOI to the datasets must be included in the Data Accessibility section of the article

(<https://royalsociety.org/journals/ethics-policies/data-sharing-mining/>). Reference(s) to datasets should also be included in the reference list of the article with DOIs (where available).

If you wish to submit your data to Dryad (<http://datadryad.org/>) and have not already done so you can submit your data via this link [http://datadryad.org/submit?journalID=RSPB&manu=\(Document not available\)](http://datadryad.org/submit?journalID=RSPB&manu=(Document%20not%20available)), which will take you to your unique entry in the Dryad repository.

Please submit a copy of your revised paper within three weeks. If we do not hear from you within this time your manuscript will be rejected. If you are unable to meet this deadline please let us know as soon as possible, as we may be able to grant a short extension.

Best wishes,
Professor John Hutchinson
mailto: proceedingsb@royalsociety.org

Associate Editor

Comments to Author:

This is a well-written article that highlights and summarizes the current problematic use of metrics, such as the h-index and impact factor. It also outlines their potentially even more disastrous impact on science in the near future - at least if we do not instil any changes very soon. The authors make a clear urge towards senior faculty to take action and the lead to reorganise the system from within academia, highlighting scientific quality and quality in mentoring as the main factors for evaluation.

While the problematic consequences of metrics per se have been highlighted frequently (as also noted by both reviewers), the novelty in the paper lies in its nuanced connections of the usage of such metrics to scientific quality in practice and which types of scientists are kept in the system. It

is important that this paper has been written by a wide range of nationality of authors, and I agree with reviewer #2 that this could be highlighted more distinctly so.

Please revise the paper according to the suggested changes made by the two reviewers, who have both recommended that the paper should be accepted with minor revisions. In particular, I agree that a couple of discussions should be expanded upon and highlighted more clearly (DORA, Metric's Tide, and I would also recommend adding the "Leiden manifesto"; D Hicks, P Wouters, L Waltman, S De Rijcke, I Rafols 2015). Both reviewers also highlighted some problems with the figure provided – could you please clarify the figure legend as well as add some discussion concerning the complication that reviewer #1 points out)? The authors should also discuss the article mentioned by reviewer # 1 and how it fits into the debate (Eyre-Walker and Stoletzki 2013). It would also be interesting to discuss whether and how funders could make useful changes to the evaluation system (this is indeed a very timely debate, as also both reviewers point out).

I hope the suggested changes can easily be made, and I am looking forward to see the revised manuscript!

Reviewer(s)' Comments to Author:

Referee: 1

Comments to the Author(s)

The topic of this MS is well-suited to the new 'Biological Science Practices' section of the journal. It deals with the ways in which scientists and institutions have reacted, and continue to react to the use of bibliometric indices in assessment of quality. The piece is very well written. It paints quite a bleak picture of widespread 'gaming' of the system and argues that this is detrimental to the quality of scientific output and to the careers of many promising young scientists.

The arguments here are familiar and most of the article is devoted to reviewing multiple previous studies on the operation and failures of indices like the Journal Impact Factor and the h-index.

The only new data come from a survey of trends in numbers of authors per article in leading specialist and generalist journals. The upward trends are worrying but the authors are unable to separate legitimate causes (such as increasing collaborative working across teams) from gaming by inappropriate addition of authors.

There is no reference to Eyre-Walker and Stoletzki (2013). I think this would be a useful paper to consider because of its comparison between expert assessment of the quality of papers and assessment based on indices. It actually questions the assumption that expert assessment is preferable, by showing that the correlation between assessments by different experts is low. This is relevant to the recommendation to that more time and effort should go into critical reading of output, rather than relying on indices, for example to compare applicants for positions.

The authors conclude by placing the burden on senior faculty to change their own behaviour and instil better attitudes in the young scientists they train. Such a change would certainly be a good thing but one must keep in mind that senior faculty hold their current positions through their success in playing the game: they may be the least likely group to see a strong motivation for changing behaviour. On the other hand, there is strong evidence that institutions (particularly funding bodies) do have the power to change behaviour. This is very clear from the way UK academics and universities have adjusted their priorities in response to the RAE/REF. This has not always pushed in the right direction but the push has certainly been effective.

I see this MS as a worthwhile contribution to an ongoing debate. Most readers will already be aware of the main points raised: this is not a new discussion. The new data confirm a trend that is familiar anecdotally and they form quite a small part of the MS. Perhaps the conclusion, with its call for action, is the most valuable element.

Eyre-Walker, Adam and Stoletzki, Nina (2013) The assessment of science: the relative merits of post-publication review, the impact factor, and the number of citations. *PLoS Biology*, 11 (10). e1001675. ISSN 1544-9173

Referee: 2

Comments to the Author(s)

This is an interesting and informative discussion of the well-known problem in modern research of gaming publication metrics. It is not particularly original – this is well-trodden ground – but there seems to be value in laying these arguments before the Ecology and Evolution community, who appear to be the primary target.

The article provides an interesting and well-structured discussion of the main issues. In a few places there seemed to me to be some gaps in consideration of the literature but that perhaps reflects my UK-centric perspective. I think this is a useful contribution to an important and growing discussion.

I have some minor comments that I think the authors should consider. I have followed the authors' page numbering.

Page 2: "much of this competition now centers on the ability of researchers to publish in high quality journals." I would question the use of the term 'high quality' in this context. Certainly, there is pressure to publish in journals with high impact factors and/or established brands, but these terms have a complicated relationship with 'quality'. Arguably there are some very high-quality disciplinary journals that do not have very high impact factors. Perhaps this could be explored?

Page 3: the statement "an extensive body of literature" is supported by a single reference to a one-page editorial in *Science*. I think the authors could do more to cite the primary literature selected from this body of evidence.

Page 4: One of the strengths of the paper is the wide range of nationalities of the authors. I think it would be helpful to readers to list these in the body of the text so it is clear what geographical contexts/experiences are being discussed.

Page 5 (and elsewhere): reference is made to the "h factor" but there is no such thing. The metric in question is the "h index".

Page 7: Some of the difficulties associated with the use of alternative metrics were discussed in some depth in *The Metric Tide* report (2015). See especially the supplementary literature review: <https://re.ukri.org/news-events-publications/publications/metric-tide/>

Page 8: Rather oblique reference is made to the Declaration on Research Assessment (DORA). This is incorrectly described as an initiative of the American Society for Cell Biology. Although the ASCB were certainly involved, so was EMBO and so were other stakeholders. More up-to-date information on the number of organisational signatories (now over 1500 – not just 75!) can be found at <https://sfdora.org> (e.g. see <https://sfdora.org/signers/>)

Page 8: Publisher profit margins are reported as being "as high as 26%" but I have seen even higher reports (more like 35%) e.g.

<https://www.theguardian.com/science/2017/jun/27/profitable-business-scientific-publishing-bad-for-science>; <https://journals.uic.edu/ojs/index.php/fm/article/view/4370/3685doi:10.5210>

Page 9: "we see a critical need to encourage a more proactive discussion". I agree but think it may be useful to discuss recent initiatives by the Wellcome Trust and the linkage in Plan S of the drive for open access with necessary reform of research evaluation.

Page 9: The question of researchers' ethics is raised here for the first time (despite 'ethics' being a keyword for the article). But the issue – and I agree it is an important one – is barely discussed. There is a fundamental disconnect, I think, between the ethical impulses that bring many people into a career in research and the mechanisms by which they are evaluated. This seems to me to be

one of the root causes of the increasing concerns about reproducibility. I think this topic could be explored in more detail – but perhaps it is not such a big issue in ecology and evolution?

Legend to Fig. 2: It would be helpful to explain what the 0-1 (or is it 0-100?) scale means. I am not familiar with Hofstede's cultural dimensions theory.

Author's Response to Decision Letter for (RSPB-2019-2047.R0)

See Appendix A.

Decision letter (RSPB-2019-2047.R1)

11-Nov-2019

Dear Dr Chapman

I am pleased to inform you that your manuscript entitled "Games academics play and their consequences: How authorship, h-index, and journal impact factors are shaping the future of academia" has been accepted for publication in Proceedings B. Congratulations!!

Open Access

Paper charges

Sincerely,

Professor John Hutchinson

Associate Editor:

Board Member

Comments to Author:

Thanks so much for taking all the reviewer comments so seriously. I think you have done a very good job incorporating all the suggested changes, so I am happy to recommend accepting the manuscript in its current format.

Appendix A

Dear Professor John Hutchinson:

Thank you for handling our submission “Games academics play and their consequences: How authorship, h-index, and journal impact factors are shaping the future of academia” and for getting reviewers who took such time and care in constructing their thoughtful reviews. We thank the reviewers for pointing out a couple of very important issues that our field needs to consider with respect to the topic of our paper.

In our response, we will deal with each reviewer’s comments one by one. We hope that our comments and responses fully meet your satisfaction and we are looking forward to working with you on our manuscript in the future.

We have listed the reviewer’s comments and our response and modification to them immediately following the comment – often inserting the text that we have changed. If multiple suggestions are made within a single paragraph, the paragraph has been broken down into each sentence/point that raises a different issue. We hope this makes things clear and easy for you when you evaluate our revisions. As instructed, we have included a track changes version and a new clean version. We made several small editorial changes throughout the manuscript to improve the clarity of our work and deleted some minor text to shorten the manuscript – these were not suggested changes, but we hope they are helpful.

Thanks for handling the submission.

All the best

Colin and co-authors.

Associate Editor - Professor John Hutchinson

Comment 1:

This is a well-written article that highlights and summarizes the current problematic use of metrics, such as the h-index and impact factor. It also outlines their potentially even more disastrous impact on science in the near future - at least if we do not instill any changes very soon. The authors make a clear urge towards senior faculty to take action and the lead to reorganize the system from within academia, highlighting scientific quality and quality in mentoring as the main factors for evaluation.

While the problematic consequences of metrics per se have been highlighted frequently (as also noted by both reviewers), the novelty in the paper lies in its nuanced connections of the usage of such metrics to scientific quality in practice and which types of scientists are kept in the system. It is important that this paper has been written by a wide range of nationality of authors, and I agree with reviewer #2 that this could be highlighted more distinctly so.

Response:

We were pleased that the Associate Editor viewed the article well written and highlighted what we thought was one of our most important contributions; namely the potential for “the

current problematic use of metrics, their potentially even more disastrous impact on science in the near future”.

Comment 2:

Please revise the paper according to the suggested changes made by the two reviewers, who have both recommended that the paper should be accepted with minor revisions. In particular, I agree that a couple of discussions should be expanded upon and highlighted more clearly (DORA, Metric’s Tide, and I would also recommend adding the “Leiden manifesto”; D Hicks, P Wouters, L Waltman, S De Rijcke, I Rafols 2015).

Response:

As suggested, we have expanded on our presentation of DORA, Metric’s Tide, and Leiden manifesto. We had done this immediately preceding our objective statement to highlight its importance. We have inserted the following text:

“Calls to mobilize reform in the tools, practices, and study of the assessment of science and scientists gained momentum this decade, particularly through the San Francisco Declaration on Research Assessment (DORA[18]). Published in 2013 and now signed by some 15,000 individuals and 1550 organizations worldwide, DORA recommends that journal impact factor not be used as a surrogate measure of an individual research article, to assess an individual scientist’s contribution, or in hiring, promotion, or funding decisions. Steps that individuals, universities, funders, and scientific societies can take to stop the mismeasure of the quality of research output are illustrated by a growing list of good practices, such as the French National Research Agency’s awareness and training program announced with the 2019 call for proposals. Use of improved indicators by all stakeholders in the system is echoed in the ten principles of the 2015 Leiden Manifesto, which includes the role of qualitative judgement and contextualization of metrics for the research field, and other international or national efforts [19]. The Metric Tide report (2015) makes twenty recommendations specific to improve research assessment in the U.K. system; however, the dangers raised that journal impact factor and citation counts can be gamed are universal [20]. More recently after reviewing 22 documents impacting clinicians and life scientists, Moher and colleagues [21] outline six general principles about what to assess, how to assess it, the need for complete and transparent publication along with openness of data and results, the need for research on new and existing assessment criteria, and the importance of rewarding intellectual risk-taking to encourage ground-breaking research. Despite the groundswell of calls for transformation of the assessments and incentive structures, systemic change is difficult and slow. Implementation of any of these recommendations and principles will be out of pace with researchers’ current career path.”

Comment 3:

Both reviewers also highlighted some problems with the figure provided – could you please clarify the figure legend as well as add some discussion concerning the complication that Reviewer #1 points out)?

Response:

As Reviewer #1 suggests, we added to the text where Figure 1 is first mentioned to clarify the issue that the increased numbers of authors on publications can represent either the increased multidisciplinary of science or an attempt to game the system. The have now included the following text:

The trend for an increased number of authors on papers is evident in all types of journals, be the journal regional, taxonomic, theoretical to a subfield, or the most general scientific journal (Fig. 1). This partially represents science becoming more multi-disciplinary and that researchers with different skills and tools are needed to address the question of interest. However, the multi-author trend also facilitates developments that are not so positive; namely including authors to game the system to increase the number of papers an author publishes.

As suggested by Reviewer #2, for Figure 2 we have provided a more detailed description of what the individualistic score represents. The figure legend now reads:

Fig 2. The individualistic score (based on the Hofstede's cultural dimensions theory – this is the amount to which people in a society/cultural are integrated into groups) of countries (<https://www.hofstede-insights.com/product/compare-countries/>) for which scores are available. The higher this score the greater the preference for a loosely-knit social framework in which individuals are expected to take care of only themselves and their immediate families. While the lower the score, the greater preference for a tightly-knit framework in society where people expect their relatives or members of a particular group to look after them in exchange for unquestioning loyalty. Countries in grey do not have available data. Countries with a lower score are thought to be more likely to include unmerited authors on publications [22]. Countries in grey do not have available data.

Comment 4:

The authors should also discuss the article mentioned by reviewer # 1 and how it fits into the debate (Eyre-Walker and Stoletzki 2013).

Response:

To respond to the reviewer's suggestion, we have read this paper, a number of papers that the paper cites, and papers that have cited this work. We have commented on the ideas of this paper in detail, particularly with reference to the assessment of the value of expert review.

The following has been inserted.

With increasing demands being placed on universities, granting agencies, and researchers' limited time, there will be an increased pressure to use shortcuts, such as metrics, rather than investing in expert review [43]. In fact, some granting agencies (e.g., National Geographic) have scaled back on their reliance on expert review as costs mount and submissions increase. A recent analysis by Eyre-Walker and Stoletzki [44] suggests that post-publication peer review is prone to error, biased by reviewer's perception of journal impact factor, and expensive. This leads them to question the use of expensive expert review in post-review assessments and to suggest journal impact factor is more appropriate. They find a lack of agreement among reviewers and interpret this to indicate that these assessments are not reliable. However, the authors fail to recognize that this lack of agreement may reflect that reviewers often assess different aspect of research or may be considering a work's merit to different sub-disciplines [45]. The pressure to use metrics as short-cuts and the recognition that each metric has its limitation have called for assessments using of multiple metrics (e.g., number of views, researcher bookmarking, social media) [45] and these metrics, particularly social media, have been promoted by publisher and Universities. Altmetric is a one such metric that is gaining prominence. It provides a score of the online attention received by research outputs based on social media (e.g., Twitter and Facebook), traditional media, blogs (both from institutions and individuals), and online reference managers. While improving metrics may be helpful to evaluate some aspects of research life, we strongly recommend that they should not be rewarded with perverse incentives, nor should the metric be easily gamed. Altmetric scores can be gamed by posting more on social media and having your friends post or even writing programs to post or repeatedly download your articles. Furthermore, rewarding individuals with high scores on such metrics selects for particular types of scientists, takes time away from research. The use of social media in academia must be made with caution so that measuring the traces of research impact does not become the goal, rather than the quality of the research itself [8]. We encourage rather than developing other metrics, that academia improves systems of expert review to ensure quality. This could include have reviews shared among evaluators and receive evaluators joint comments before deciding on publications or grants and having reviewing receiving more importance in the tenure and promotion process.

Comment 5:

It would also be interesting to discuss whether and how funders could make useful changes to the evaluation system (this is indeed a very timely debate, as also both reviewers point out).

Response:

Authors on this paper are/or were program directors or on review panels of several important granting agencies, such as the French National Research Agency (France), Deutsche Forschungsgemeinschaft (Germany) National Science Foundation (USA), National Geographic Society (USA), and the National Science Foundation of China. Based on this experience, we have added the following text.

Funders must also play a leading role in changing academic culture with respect to how the game is played. First and foremost, funders have a clear role in setting professional and ethical standards. For example, they can outline the appropriate standards in the treatment of colleagues and students with respect to such difficult questions as what warrants authorship and how to determine its ordering. Granting agencies should clearly emphasize the importance of quality and send a clear message that indices should not be used, as is expressed by DORA that many agencies have endorsed. Of particular importance, is for funders not to monetize research outputs based on metrics, such as the h-index or journal impact factor. Monetization largely based on such metrics is being done in many countries, such as Australia, China, Mexico, Scandinavia, South Africa, and Uganda, and incentives can be as high as \$165,000 US per publication [19, 47].

Referee: 1

Comment 6 (numbering continuing from above):

Comments to the Author(s)

The topic of this MS is well-suited to the new 'Biological Science Practices' section of the journal. It deals with the ways in which scientists and institutions have reacted and continue to react to the use of bibliometric indices in assessment of quality. The piece is very well written. It paints quite a bleak picture of widespread 'gaming' of the system and argues that this is detrimental to the quality of scientific output and to the careers of many promising young scientists.

Response:

We are glad the reviewer viewed the piece to be very well written and are pleased to see she/he grasped the important issues we were raising.

Comment 7:

The arguments here are familiar and most of the article is devoted to reviewing multiple previous studies on the operation and failures of indices like the Journal Impact Factor and the h-index. The only new data come from a survey of trends in numbers of authors per article in leading specialist and generalist journals. The upward trends are worrying but the authors are unable to separate legitimate causes (such as increasing collaborative working across teams) from gaming by inappropriate addition of authors.

Response:

We agree with these comments, but no response is required.

Comment 8:

There is no reference to Eyre-Walker and Stoletzki (2013). I think this would be a useful paper to consider because of its comparison between expert assessment of the quality of papers and assessment based on indices. It actually questions the assumption that expert assessment is preferable, by showing that the correlation between assessments by different experts is low. This

is relevant to the recommendation to that more time and effort should go into critical reading of output, rather than relying on indices, for example to compare applicants for positions.

Response:

To respond to the reviewer's suggestion, we have read this paper, several papers that the paper cites, and papers that have cited this work. We have commented on the ideas of this paper in detail. See our response above and the included modified text to where the Associate Editor raised this issue.

Comment 9:

The authors conclude by placing the burden on senior faculty to change their own behaviour and instil better attitudes in the young scientists they train. Such a change would certainly be a good thing but one must keep in mind that senior faculty hold their current positions through their success in playing the game: they may be the least likely group to see a strong motivation for changing behaviour.

Response:

Good point. We have added the following text to the conclusions where we discuss the role of senior faculty.

One must keep in mind that senior faculty likely hold their current positions through their success in the game, which may or may not have been achieved by using the most ethical ways. Thus, institutions must also play a role by training mentors to create a healthy and ethically robust culture and encouraging team mentorship where deviations from the highest professional standards can be monitored and, where necessary, appropriately disciplined.

Comment 10:

On the other hand, there is strong evidence that institutions (particularly funding bodies) do have the power to change behaviour. This is very clear from the way UK academics and universities have adjusted their priorities in response to the RAE/REF. This has not always pushed in the right direction but the push has certainly been effective.

Response:

See response to Comment 5 above.

I see this MS as a worthwhile contribution to an ongoing debate. Most readers will already be aware of the main points raised: this is not a new discussion. The new data confirm a trend that is familiar anecdotally and they form quite a small part of the MS. Perhaps the conclusion, with its call for action, is the most valuable element.

Response:

We agree with this comment, but no response is needed.

Referee: 2

Comment 11:

This is an interesting and informative discussion of the well-known problem in modern research of gaming publication metrics. It is not particularly original – this is well-trodden ground – but there seems to be value in laying these arguments before the Ecology and Evolution community, who appear to be the primary target.

The article provides an interesting and well-structured discussion of the main issues. In a few places there seemed to me to be some gaps in consideration of the literature but that perhaps reflects my UK-centric perspective. I think this is a useful contribution to an important and growing discussion.

Response:

We thank the reviewer for the kind comments, and we have attempted to fill in the gaps in the literature as much as possible, while at the same time considering the reference limitation and page length of this type of article.

Comment 12:

Page 2: “much of this competition now centers on the ability of researchers to publish in high quality journals.” I would question the use of the term ‘high quality’ in this context. Certainly, there is pressure to publish in journals with high impact factors and/or established brands, but these terms have a complicated relationship with ‘quality’. Arguably there are some very high-quality disciplinary journals that do not have very high impact factors. Perhaps this could be explored?

Response:

We agree with the reviewer’s comment and have changed this from high quality to journals with high impact factors.

We agree that with the idea that some high-quality disciplinary journals do not have high impact factors but are still important to the field and are publishing quality research. However, we could not think of any easy way to substantiate this idea and given page restrictions for this type of submission and the importance of other points we were asked to add, we have not explored this in the revised manuscript.

Comment 13:

Page 3: the statement “an extensive body of literature” is supported by a single reference to a one-page editorial in Science. I think the authors could do more to cite the primary literature selected from this body of evidence.

Response:

To meet the reviewer’s request, we have cited a review article on this topic. We were attempting to keep the number of references down to the most significant but adding this review article is appropriate.

Comment 14:

Page 4: One of the strengths of the paper is the wide range of nationalities of the authors. I think it would be helpful to readers to list these in the body of the text, so it is clear what geographical contexts/experiences are being discussed.

Response:

As suggested, we have listed the countries represent by the authors. Note these do not always correspond to the current affiliations of everyone.

Comment 15:

Page 5 (and elsewhere): reference is made to the “h factor” but there is no such thing. The metric in question is the “h index”.

Response:

As suggested, this has been corrected throughout.

Comment 16:

Page 7: Some of the difficulties associated with the use of altermatic metrics were discussed in some depth in The Metric Tide report (2015). See especially the supplementary literature review: <https://re.ukri.org/news-events-publications/publications/metric-tide/>

Response:

We had included a discussion of the difficulties of metrics in general and particularly altmetrics. And we have added a more detailed discussion of the Metric Tide Report – see above in the responses to the Associate Editor.

In addition, as indicated above, we have included the following new text.

The pressure to use metrics as short-cuts and the recognition that each metric has its limitation have called for assessments using of multiple metrics (e.g., number of views, researcher bookmarking, social media) [45] and these metrics, particularly social media, have been promoted by publisher and Universities. Altmetric is a one such metric that is gaining prominence. It provides a score of the online attention received by research outputs based on social media (e.g., Twitter and Facebook), traditional media, blogs (both from institutions and individuals), and online reference managers. While improving metrics may be helpful to evaluate some aspects of research life, we strongly recommend that they should not be rewarded with perverse incentives, nor should the metric be easily gamed. Altmetric scores can be gamed by posting more on social media and having your friends post or even writing programs to post or repeatedly download your articles. Furthermore, rewarding individuals with high scores on such metrics selects for particular types of scientists, takes time away from research. The use of social media in academia must be made with caution so that measuring the traces of research impact does not become the goal, rather than the quality of the research itself [8]. We encourage rather than developing other metrics, that academia improves systems of expert review to ensure quality. This could include have reviews shared among evaluators and receive evaluators joint comments before deciding on publications or grants and having reviewing receiving more importance in the tenure and promotion process.

Comment 17:

Page 8: Rather oblique reference is made to the Declaration on Research Assessment (DORA). This is incorrectly described as an initiative of the American Society for Cell Biology. Although the ASCB were certainly involved, so was EMBO and so were other stakeholders. More up-to-

date information on the number of organisational signatories (now over 1500 – not just 75!) can be found at <https://sfdora.org> (e.g. see <https://sfdora.org/signers/>)

Response:

We thank the reviewer for the updated information. We have refined this statement and provided up to date information on the number of institutional and individual signatories and associated supporting groups.

We have added more information about DORA, The Liden Manifesto, etc. and please see our response to Comment 2 above.

Comment 18:

Page 8: Publisher profit margins are reported as being “as high as 26%” but I have seen even higher reports (more like 35%) e.g. <https://www.theguardian.com/science/2017/jun/27/profitable-business-scientific-publishing-bad-for-science>; <https://journals.uic.edu/ojs/index.php/fm/article/view/4370/3685doi:10.5210>

Response:

We thank the reviewer for pointing out these references and we have now cited one in the paper, including extra information that it provided. The modified and added text is the following:

Academics should keep in mind that journals are, for the most part, a for-profit business with profit margins reaching as high as 4026% [3] [33]. In fact, in 2017 the global revenues from scientific publishing was estimated to be £19bn (~ 24bn US), and in 2010 its profit margins were higher than Apple, Google, or Amazon [33]. The need for profit can surpass doing what is best in an academic or societal sense. A major way journals manipulate their impact factor is by

Comment 19:

Page 9: “we see a critical need to encourage a more proactive discussion”. I agree but think it may be useful to discuss recent initiatives by the Wellcome Trust and the linkage in Plan S of the drive for open access with necessary reform of research evaluation.

Response:

We agree that there is a need to reform business models of publications and that of how universities are currently structured under. Neither of these models are driven by evaluating quality of science, rather they are driven too much by quantity. As previously suggested by the Associate Editor and Reviewer #1, we have discussed efforts that have been made and are currently being developed to “to encourage a more proactive discussion”. However, to go into the value of open access and Plan S would be lengthy, particularly given its controversies. Thus, we have opted to only discuss the efforts such as DORA, Leiden Manifesto, and Metric Tide.

Comment 20:

Page 9: The question of researchers’ ethics is raised here for the first time (despite ‘ethics’ being a keyword for the article). But the issue – and I agree it is an important one – is barely discussed.

There is a fundamental disconnect, I think, between the ethical impulses that bring many people into a career in research and the mechanisms by which they are evaluated. This seems to me to be one of the root causes of the increasing concerns about reproducibility. I think this topic could be explored in more detail – but perhaps it is not such a big issue in ecology and evolution?

Response:

Ethics is an important issue in ecology and evolution and there are many publications dealing with this, including ones written by the authors. As a result, we agree that ethics should be discussed more fully. Thus, we have now raised the topic of ethics several times but given the page limit of these types of articles we have not elaborated on ethics in detail in a distinct section.

Comment 21:

Legend to Fig. 2: It would be helpful to explain what the 0-1 (or is it 0-100?) scale means. I am not familiar with Hofstede's cultural dimensions theory.

Response:

See response to Comment 3 above for the new figure legend. In general, this index describes the values of members of a society and how these values relate to behavior and was derived from a factor analysis. It was originally based on four dimensions (which the reviewer appears to know), but a fifth and then a sixth were added. The results can be expressed as a percentage as we have done. One of those dimensions is Individualism and as suggested we have provided a discussion of what this represents and interested readers can easily go to the referred website.